# The Role of Acyl-CoA β-Oxidation in Brain Metabolism and Neurodegenerative Diseases

**DOI:** 10.3390/ijms241813977

**Published:** 2023-09-12

**Authors:** Sylwia Szrok-Jurga, Jacek Turyn, Areta Hebanowska, Julian Swierczynski, Aleksandra Czumaj, Tomasz Sledzinski, Ewa Stelmanska

**Affiliations:** 1Department of Biochemistry, Faculty of Medicine, Medical University of Gdansk, 80-211 Gdansk, Poland; jacek.turyn@gumed.edu.pl (J.T.); areta.hebanowska@gumed.edu.pl (A.H.); 2Institute of Nursing and Medical Rescue, State University of Applied Sciences in Koszalin, 75-582 Koszalin, Poland; juls@gumed.edu.pl; 3Department of Pharmaceutical Biochemistry, Faculty of Pharmacy, Medical University of Gdansk, 80-211 Gdansk, Poland; aleksandra.czumaj@gumed.edu.pl (A.C.); tomasz.sledzinski@gumed.edu.pl (T.S.)

**Keywords:** beta-oxidation, fatty acid metabolism, brain, neurodegenerative diseases, aging

## Abstract

This review highlights the complex role of fatty acid β-oxidation in brain metabolism. It demonstrates the fundamental importance of fatty acid degradation as a fuel in energy balance and as an essential component in lipid homeostasis, brain aging, and neurodegenerative disorders.

## 1. Introduction

Compared with the heart, kidney, or liver, there are relatively studies about fatty acid (FA) β-oxidation (βOX) in the nervous system. The value of the brain’s respiratory quotient (RQ), calculated as a proportion of produced CO_2_ and consumed O_2_, is close to 1 [1]. Therefore, it was widely accepted for decades that the brain does not oxidize FAs and that neurons mainly use glucose under physiological conditions and additionally ketone bodies (KBs) in a fasted state as an energy source. The brain is a sensitive organ, and even small environmental changes can result in serious impairment. As oxygen consumption releases a significant amount of reactive oxygen species (ROS), mainly in the form of superoxide radicals, βOX has been considered only as a detrimental aspect, potentially dangerous for neurons [2]. Current published data indicate that the energy metabolism of the central nervous system (CNS) is much more complex and compartmentalized. However, as an alternative substrate, FAs cannot fully replace glucose and reverse the behavioral consequences of hypoglycemia [3]. They are critical for proper CNS functioning. Furthermore, a significant link between brain βOX and neurodegenerative diseases has recently been proposed [4]. Since it is known that long-chain FAs (LCFAs) and medium-chain FAs (MCFAs) can cross the blood–brain barrier (BBB), [5,6] the ability of its oxidation to CO_2_ and H_2_O in astrocytes was also tested [7,8]. The in situ perfusion technique of rat brains demonstrated that saturated and unsaturated FAs are rapidly transported through the BBB [9]. The BBB is formed by tightly adherent endothelial cells of blood capillaries and astroglial cells that line the cerebral microvessels, creating a virtually continuous sheath around the vessel walls [10]. This structure forces transcellular transport for most compounds, including ions and FAs, to rely on specific transport systems. Passing FAs from the blood to the brain may occur by diffusion and/or proteins facilitating transport. The major FA transporters expressed in human brain microvessel endothelial cells, mouse capillaries, and human grey matter are FA transport proteins (FATP1, FATP4), FA binding protein 5 (FABP5), and FA translocase/CD36 [11,12]. Moreover, it has been recently demonstrated that some LCFAs (docosahexaenoic, oleic, and palmitic acids) are transported across the BBB in the form of lysophosphatidylcholine by a significant facilitator superfamily domain containing 2A (MFSD2A) in a Na^+^-dependent manner [13,14]. In addition, Panov et al. [7] revealed that the hypothalamus has relatively easy access to many low molecular weight compounds present in the blood, such as carnitine and LCFAs, through numerous “fenestrations” in the BBB. It was also suggested that FA metabolism within discrete hypothalamic regions might function as a nutrient availability sensor integrating multiple nutritional and hormonal signals [15]. Thus, it can be summarized that the available data do not support the view that the limited passage of FAs through the BBB is the cause of poor βOX by the brain. After cellular uptake, FAs are enzymatically activated to acyl-CoA derivatives. In turn, acyl-CoA can be either esterified to membrane lipids or degraded by mitochondrial βOX to provide cellular energy. βOX provides up to 20% of the brain’s energy expenditure [16]. Because astrocytes envelop microvessels in the brain, they are likely to be the primary cells of FA metabolism in the brain. Indeed, in primary cultures, only astrocytes could utilize FA for CO_2_ production, and the utilization rate was greater than that of the KBs [8]. Significant βOX activity was also detected in neural progenitor cells, which are known to form new neurons during development [17] and in the adult brain [18]. It is widely accepted that neuronal mitochondria in the adult brain do not oxidize FAs [19]. Reduced oxidative catabolism of LCFAs (C12 to C18) was observed in both homogenates of neural cells and in mitochondria obtained from rodents’ brains [19]. The low level of LCFAs oxidation by isolated brain mitochondria was explained by a decreased translocation rate of LCFA-CoA esters across the inner mitochondrial membrane and decreased enzymatic capacity of the βOX pathway. It was demonstrated that carnitine palmitoyltransferase 1 (CPT1C), the main brain-specific enzyme playing a crucial role in LCFA transport into mitochondria, and some FA oxidation enzymes have low activity [19,20]. For instance, thiolase, acyl-CoA dehydrogenase (ACDH), and enoyl-CoA-hydratase (ECH) activities in the brain were only 0.7%, 50%, and 19%, respectively, of the activities observed in the rat heart mitochondria [19]. It is worth noting that the enzymes’ glial and neuronal forms were not assayed separately. As a result, the slow βOX rate appears to be a distinctive intrinsic characteristic of brain tissue, especially in the mitochondria of neurons. Schönfeld and Reiser suggested that this phenomenon may protect neurons from the harmful effects of FA oxidation since βOX is the major source of ROS production and the antioxidant defense capacity of neurons is low. Therefore, reducing the rate of βOX prevents oxidative stress. In addition, since adenosine triphosphate (ATP) generation associated with FA oxidation (FAO) requires more oxygen than glucose oxidation, reduced βOX decreases the risk of neuronal hypoxia [21].

However, it is interesting to consider the potential role of βOX in brain metabolism, particularly in response to the fact that astrocytes oxidize FAs, and that FABPs and carnitine are present in brain tissue. Carnitine, widely known for its significant role in transporting LCFAs across the inner mitochondrial membrane, is essential for proper neurological function [22]. A lack of carnitine may lead to metabolic encephalopathy, characterized by swollen astrocytes and expanded mitochondria [23]. Jernberg et al. demonstrated that CPT1A (with higher activity than CPT1C) is exclusively present in astrocytes and neural progenitor cells, while it is absent in rats’ neurons, microglia, and oligodendrocytes [24].

Based on proteomic predictions, Fecher et al. revealed that astrocytic mitochondria metabolize LCFAs more efficiently than the neuronal mitochondria of mice [25]. Moreover, recent studies comparing mitochondrial transcriptomes and proteomes between different neural cell types confirmed the increased concentration of enzymes involved in FA metabolism and transport in astrocytes compared with neurons in mice and humans [19,26]. In the brain, 95% of ECH and hydroxy-acyl-CoA dehydrogenase co-localize with the astrocyte biomarker: glial fibrillary acidic protein [27]. Short-chain-specific acyl-CoA dehydrogenase was also predominantly found in astrocytes [25]. As revealed, astrocytes have all the machinery for metabolizing MCFAs, LCFAs, and very-long-chain fatty acids (VLCFAs), reflected not only by the higher expression of FA oxidation-related enzymes but also by the higher content of peroxisomes [25,28].

Peroxisomes are commonly found in different central nervous system cells, particularly within glial cells. Recent research has revealed that peroxisomes are present within the innermost layer of the myelin sheath [29]. βOX in these organelles primarily involves substrates that cannot be oxidized directly in the mitochondria [30]. One of the main functions of peroxisomes in the brain is the catabolism of saturated VLCFAs (mainly C24:0 and C26:0). Therefore, peroxisomes regulate the level of molecules significantly abundant in myelin [31]. Degradation of VLCFAs in peroxisomes requires several cycles of (a) dehydrogenation, (b) hydration, (c) another round of dehydrogenation, and (d) thiolytic cleavage catalyzed by acyl-CoA oxidase (ACOX), (ACOX1 is known to selectively react with CoA esters of VLCFAs, while ACOX2 is responsible for oxidizing branched-chain FAs), D-bifunctional proteins, and peroxisomal thiolase [32]. Each cycle leads to the release of a shortened FAs and acetyl-CoA. In contrast to mitochondrial FAO during the peroxisomal process, FAD is reduced by ACOX. Electrons from FADH_2_ are transported to the oxygen, generating hydrogen peroxide (H_2_O_2_). Additionally, the released FA undergoes subsequent rounds of βOX until the formation of MCFA-CoA/MCFA, which is transported to mitochondria to complete degradation (βOX) [33]. Acetyl-CoA produced during peroxisomal FAO may become a substrate in the synthesis of cholesterol and phospholipids, predominantly plasmalogen [34]. Thus, peroxisomal dysfunction can result in significant neuropathological consequences, while the absence of plasmalogens is a recognized pathological factor in Zellweger Syndrome [35]. Furthermore, increased VLCFA concentration and decreased plasmalogen levels may lead to impaired neurotransmission and accumulation of β-amyloid plaques associated with Alzheimer’s disease (AD) [36]. It was also revealed that an increased level of VLCFAs leads to the degeneration of astrocytes and oligodendrocytes. These disturbances cause an imbalance in the regulation of intracellular Ca^2+^ and a notable decrease in the membrane potential of mitochondria in oligodendrocytes, particularly in individuals who have X-linked adrenoleukodystrophy (X-ALD) and ACOX1 deficiency [37]. Moreover, peroxisomal oxidation is essential for the catabolism of monounsaturated, polyunsaturated FAs and their derivatives (eicosanoids and docosanoids), 2-methyl branched-chain FAs, and dicarboxylic acids. Because of the differences in substrate composition, the degradation of unsaturated and branched FAs in peroxisomes involves several multifunctional enzyme complexes. The catabolism of eicosanoids is particularly important for maintaining the appropriate levels of arachidonic acid, eicosapentaenoic acid, and products of their metabolism, such as prostaglandins, leukotrienes, prostacyclins, and thromboxanes [30]. Therefore, disturbances in peroxisomal FAO may contribute to the brain’s inflammatory processes. It is worth remembering that the α-oxidation of phytanic acid (3,7,11,15-tetramethyl hexadecanoic acid) in peroxisomes is especially crucial in the brain. A deficiency of phytanoyl-CoA 2-hydroxylase (converting phytanic acid to pristinic acid) results in neurological damage in the form of Refsume disease [38].

The oxidation of FAs also plays a vital role in the myelinating cells of the CNS, oligodendrocytes, where it participates in lipid turnover, providing essential elements for the synthesis of new myelin lipids [39,40]. In addition, under conditions of significant energy restriction, oligodendrocytes assist axons by metabolizing myelin lipids via βOX and production of KBs. Studies conducted on a mouse optic nerve model demonstrated that generating ATP from lipids allows for greater allocation of glucose-derived metabolites to the axons, thereby preserving them [41]. Increased FAO levels were demonstrated in microglia during glucose deprivation. Moreover, since activated microglia require more ATP, FA oxidation may partially supply this energy demand [42]. Additionally, it was also revealed that FAs act as signaling molecules affecting the activity of microglia. During the aging process in mice, it has been observed that microglia undergo changes in their energy metabolism, with a shift from glycolysis to FAO as the primary energy source [43]

## 2. Different Roles of Astrocytic βOX in Brain Metabolism

### 2.1. βOX as a Source of Energy

The main functions of the astroglia are metabolic support of neurons with nutrients such as lactate and KBs, lipids production and release, storing glycogen, neurotransmitter uptake and release, neurotransmitter synthesis de novo, ion homeostasis, the elimination of oxidative stress, tissue repair and participating in synaptic formation [44]. Thus, astroglial mitochondria must constantly produce much energy to support these functions. As in other organs, FAO in astrocytes provides NADH and FADH_2_, subsequently used in oxidative phosphorylation (OXPHOS) for ATP production. Moreover, Panov et al. suggested that βOX of FAs may be the astrocytes’ primary source of acetyl-CoA for the TCA (tricarboxylic acid cycle, Krebs cycle). This is supported by the direction of glucose metabolism and the activities of mitochondrial enzymes involved in the metabolism of pyruvate formed from glucose. About 60% of the glucose consumed in the aerobic condition is converted in astrocytes to lactate (Warburg effect) [45], constituting an excellent oxidative substrate for neurons (astrocyte–neuron lactate shuttle). Pyruvate, formed from glucose, is converted to acetyl-CoA or oxaloacetate (OAA) in the mitochondria by pyruvate dehydrogenase complex (PDC) and pyruvate carboxylase, respectively [7]. Acetyl-CoA formed during βOX inhibits PDC activity and activates pyruvate carboxylase, increasing the pool of OAA needed to incorporate acetyl-CoA into TCA [7]. Together, these properties of the critical enzymes in astrocytes direct more pyruvate to the formation of lactate and OAA (Figure 1). This means that upon neuronal activation, astrocytes simultaneously supply neurons with lactate as a precursor of pyruvate, a substrate for neuronal mitochondria. In contrast, astrocytic mitochondria provide high levels of ATP from βOX to maintain the glutamate/glutamine cycle, glycogen synthesis, and other ATP-consuming functions. Interestingly, the astrocyte-to-neuron lactate shuttle was more functional in young hippocampi, whereas aged neurons become independent of astrocytic lactate, disrupting the metabolic crosstalk between the brain’s cells [46].

### 2.2. Astrocyte βOX as a Source of Ketone Bodies

During prolonged fasting, KB levels rise significantly and can contribute almost 60% of the brain’s energy requirement, thereby replacing glucose as the primary fuel. Although it is generally known that the liver supplies the brain with KBs, some studies demonstrate that astrocytes are also ketogenic cells [47]. In astrocytes, βOX produces acetyl-CoA, which is then used to synthesize D-β-hydroxybutyrate (BHB) and released to neighboring neurons [48]. KBs, synthesized from FAs in the astrocytes, are transported to the neurons by a monocarboxylate transporter for energy production via the TCA. Such central production of KBs, in combination with the synthesis of lactate in astrocytes, may cover part of the energy need of nearby activated neurons [49] (Figure 2). As in the liver, ketogenesis in astroglia is regulated by AMP-dependent kinase (AMPK) [50]. AMPK may be stimulated by 5-aminoimidazole-4-carboxamide ribonucleotide (AICAR) [51] and metformin [52], which is a potential target for pharmacological interventions. Moreover, hypoxia and/or hypoglycemia have been shown to enhance astroglial KB production in vitro [53]. As astrocytes respond to ischemia in vitro by enhancing KB production via AMPK, it might be an essential energy source instead of lactate as PDC is susceptible to ischemia [54]. There is an opinion that astrocyte ketogenesis is a cytoprotective pathway because KBs may function as direct antioxidants, suppressing mitochondrial ROS production and promoting transcriptional activity of the antioxidant defense [55]. Fortier et al. demonstrated that MCFA supplementation (which crosses the BBB) could enhance cerebral KB metabolism and improve several cognitive outcomes in patients with mild cognitive impairments [56]. The neuroprotection by KBs is well established and clinically used as a high-ketogenic diet for epilepsy and brain injury [53,54]. These and other results thus point to the therapeutic role of the exo- and endogenic KBs in neurodegenerative disease [57]. Moreover, in addition to its role as a fuel substrate, KBs also provide an important element for synthesizing brain lipids, including myelin [58]. BHB significantly increased myelination and reduced axonal degeneration caused by glucose deprivation in an in vitro model of myelination. BHB prolonged cell viability, suppressed glucose deprivation-induced collapse of mitochondrial membrane potential, and reduced oligodendrocyte death. Accordingly, it is proposed that KBs may protect the myelin-forming oligodendrocytes and reduce axonal damage [59].

### 2.3. The Role of βOX in the Oxidation of Fatty Acids Transported from Neurons to Astrocytes

The Glial ability to deliver lipids to neurons via lipoprotein particles is well known. However, the flow of lipids, namely FAs, from neurons to astrocytes has only recently been discovered [60]. It mainly occurs in response to ROS, which induce peroxidation of FAs, and is associated with the accumulation of lipid droplets (LDs) in neighboring astrocytes and microglia in the adult mouse brain [61]. Unlike neurons, astrocytes readily make LDs and produce many antioxidants, allowing them to manage oxidative stress effectively [62]. RNA sequencing analysis showed higher expression of genes related to oxidative stress and lipid metabolism in astrocytes than in neurons. In particular, the expression of genes related to neutralizing superoxide radicals and those responsible for protection against free FA toxicity were significantly elevated in LD-containing astrocytes [63]. Excessive amounts of FAs (which may be peroxidized, disrupt mitochondrial membrane integrity, and increase the production of ROS) are produced in overactive neurons, transferred to ApoE-containing lipid particles, and then secreted into the intercellular space. Astrocytes endocytose ApoE-positive lipid particles, and released FAs are esterified and stored in LDs. The free fatty acids (FFAs) released from the LDs are then oxidized in the mitochondria (Figure 3).

Qi et al. suggested [64] that compromised astrocytic degradation of FAs could constitute a potential mechanism underlying lipid dysregulation in the brain. Each ApoE4 allele increases AD risk by 3- to 4-fold, while lowering the age at the onset by ~8 years [65]. In addition to its regulatory function in amyloid-β aggregation and clearance, ApoE4 is also significantly involved in the development of multiple neurological disorders, such as dementia with Lewy bodies, multiple sclerosis, and cerebrovascular disease [66]. ApoE4 knock-in human astrocytes resulted in a metabolic shift toward enhanced glucose metabolism and decreased βOX, with subsequent lipid accumulation in astrocytes. Furthermore, the unique role of astrocytes in brain FA degradation was confirmed in another study, revealing decreased FAO and increased triacylglycerol (TAG) accumulation in the hippocampus of young ApoE4 knock-in mice [67]. In contrast, ApoE3 promoted astrocyte-induced clearance of neuronal LDs, which suggests its significance in neuronal lipid degradation [65]. Therefore, DNA and RNA editing that leads to the conversion of the ApoE4 isoform into ApoE3 might be a potential strategy to improve brain energetics and lipid homeostasis in neurodegenerative diseases [68,69]. Ioannou et al. showed that under conditions of neuronal hyperactivity, astrocytes consume neuron-derived FAs in the mitochondria via βOX, which is associated with reduced levels of ROS and lipid peroxidation in astrocytes [61]. Thus, to protect neurons from FA-induced lipotoxicity and to provide energy in specific situations, the storage and oxidation processes of FAs appear to depend on a close metabolic association between neurons and astrocytes [63]. Moreover, a tight metabolic link between neurons and astrocytes causes metabolites to transfer to neurons to synthesize antioxidants [21]. Interestingly, Mi et al. demonstrated the significance of a functional FA catabolic machinery in astrocytes for maintaining critical neuronal support, thus indicating the fundamental role of the astrocyte as the major recipient/degrader of excess neuronal lipids. Neurons exposed to FA degradation-deficient astrocytes had reduced glycolytic and mitochondrial metabolism and reduced synaptic density. They could also not sustain synaptic activity [4]. In stable conditions, astrocytic mitochondria metabolize FAs more efficiently than neurons [25]. However, if astrocytic lipid clearance is defective, neurons’ mRNA levels of peroxisome proliferator-activated receptor α (PPARα), CPT1, CPT2, and βOX genes significantly increase. Therefore, it was suggested that in the case of reduced astrocytic FA degradation, neurons initially try to degrade FAs in the mitochondria by an adaptive activation of βOX. However, this process cannot generate the required energy. Moreover, the limited antioxidant capacity of neurons leads to increased ROS, oxidative stress, and accumulation of peroxidized lipids. In addition, excessive amounts of saturated FAs within astrocytes could potentially be released via LDs that may be absorbed by neurons, resulting in further damage to neuronal cells [4,70]. In summary, this coupling of lipid metabolism between neurons and astrocytes protects neurons from ROS and FA toxicity. It is worth noting that under nutrient-deficient conditions, the survival of astrocytes depends on LD-fueled FAO [71].

### 2.4. βOX as a Source of Carbons for Neurotransmitter Synthesis

βOX in astroglial mitochondria provides energy and part of the carbon skeleton to synthesize neuro-mediators. It mainly concerns the Glutamate/GABA (gamma-aminobutyric acid)–glutamine cycle, in which astrocyte absorption removes the two neurotransmitters from the synaptic cleft and transforms them into glutamine. The newly synthesized glutamine is transferred into neurons to resynthesize neurotransmitters. Two enzymes are mainly involved in this cycle. Glutamine synthetase, expressed only in astrocytes, uses ATP to transform glutamate into glutamine [72]. Glutamine is converted into glutamate by phosphate-activated glutaminase, and its expression in neurons is much higher than in astrocytes [44]. In addition, GABA is synthesized by glutamic acid decarboxylases, which are widely present in GABAergic neurons [73]. It should be added that the glutamate/GABA–glutamine cycle is only a deliberate simplification because it does not use exogenous glutamate. In astrocytes, glutamate is also synthesized from α-ketoglutarate by glutamate dehydrogenase, especially in elevated ammonium ion concentrations (α-ketoglutarate, an intermediate of TCA, contains carbons derived from OAA (formed from glucose) and acetyl-CoA (formed from FA)) (Figure 4).

### 2.5. βOX as a Source of Acetyl-CoA for Acetylation of Proteins

During βOX, β-keto thiolase catalyzes a reaction that produces one molecule of acetyl-CoA and acyl-CoA which is shorter by two carbons than the initial substrate. The process may be repeated until all the even-chain acyl-CoA is degraded to acetyl-CoA. Mitochondrial acetyl-CoA is converted into citrate, one of the intermediates in the TCA. However, in some circumstances (e.g., FA overload or defect of oxidative phosphorylation), it may also be exported from mitochondria, regenerated in the cytosol or nucleus by ATP-citrate lyase (ACLY), and used for the acetylation of proteins [4]. Protein acetylation is a highly specific post-translational modification that significantly affects protein functions, including gene transcription and signal transduction. The process of protein acetylation depends mainly on lysine acetyltransferases and lysine deacetylases [74].

In the latest research, Mi et al. demonstrated a crucial role of FA degradation by astrocytic OXPHOS in preventing pathological change development. Using a mouse model of astrocyte-specific OXPHOS deficiency, the researchers revealed that elevated acetyl-CoA levels (caused by an imbalance between FA load and metabolism of acetyl-CoA by TCA and OXPHOS) promote the acetylation and activation of STAT3, a mediator of astrocyte reactivity, which leads to astrocyte reactivity and the release of proinflammatory factors. At the level of the intercellular interactions, lipid-laden reactive astrocytes stimulated neuronal FAO and oxidative stress, activated microglia through interleukin-3 signaling and suppressed the biosynthesis of lipids essential for oligodendrocyte-mediated myelin turnover [4] (Figure 5).

## 3. βOX in Neurodegenerative Diseases and Aging

In neurodegenerative diseases, bioenergetics demand increases and relies upon βOX as an energy source. Thus, impaired mitochondrial βOX may play an essential role in neurodegeneration [75] and may be involved in the pathogenesis of some human disorders [76]. The existence of developmental delays and behavioral disorders associated with autism has been reported in genetic deficits in very long-chain acyl-CoA dehydrogenase [77] or long-chain 3-hydroxy acyl-CoA dehydrogenase [78]. Metabolomics studies conducted on several cohorts of children with autism spectrum disorders have also identified a subpopulation of patients with abnormally high plasma acylcarnitine levels, which may reflect a partial deficiency of βOX and/or the respiratory chain in these patients [79]. Impaired FAO was also observed in astrocytes derived from induced pluripotent stem cells of patients with AD [80]. This study showed that a synthetic PPARβ/δ agonist alleviates AD-related deficits by increasing astrocyte βOX and improving cognition in a transgenic mouse model of AD. The efficacy of PPAR antagonists that transcriptionally repress βOX genes has been recently confirmed [81]. βOX enzymes are present and active within glioma tissues [82]. Thus, pharmacological modulation of FA oxidation has been suggested as a therapeutic approach against glioblastoma cells. Mice treated with etomoxir, an inhibitor of CPT1 and thus βOX, demonstrated reduced CNS inflammation and demyelination in an experimental autoimmune encephalomyelitis model, an animal model of multiple sclerosis [83].

βOX disorders may also be linked to the development of Parkinson’s disease (PD) and Huntington’s disease (HD). It has been recently shown that mitochondrial metabolism, βOX, and mitochondrial respiration are simultaneously affected before neuronal loss and α-synuclein fibril deposition in early-stage PD [84]. An animal study revealed that glycolytic striatal astrocytes had reprogrammed their fuel use and switched to FAO as an alternative energy source in HD [85]. Moreover, behavioral and transcriptional analyses of the HD *Drosophila* model demonstrated that the availability of lipids for βOX plays an essential role in developing HD symptoms [86]. In this study, Teglicar, a reversible inhibitor of CPT1A (also known as ST1326), impaired the oxidation of FAs and the availability of cytosolic acetyl-CoA. Therefore, decreased cell reliance on βOX led to a metabolic shift toward glucose oxidation and significantly improved locomotion. It delayed the onset of the disease in the HD *Drosophila* model [86].

It is worth adding that Sayre et al. demonstrated the neuroprotective role of βOX in the brain after an ischemic stroke [27].

Stress granules (SG) are aggregates of proteins and RNA molecules that usually accumulate reversibly in response to cellular stress. They may also promote the development of toxic protein aggregates, such as those seen during the progression of certain neurological diseases [87]. Amen and Kaganovich reported that SG formation leads to a downregulation of βOX by modulating mitochondrial voltage-dependent anion channels, which import FAs into mitochondria. Interestingly, cells derived from amyotrophic lateral sclerosis (ALS) patients fail to form functional SG, leading to uncontrolled fatty acid oxidation [88]. ALS is a progressive neurodegenerative disorder associated with the selective degeneration of upper motor neurons in the motor cortex and lower motor neurons in the brainstem and spinal cord. Dysfunctions in lipid metabolism and function are potential pathogenesis drivers [2]. The lack of SG formation, which translates into an inability to lower βOX, is a new significant point that may significantly impact the development of neurodegenerative diseases [88]. An increasing amount of evidence suggests that gut microbiota are involved in the bidirectional communication between the gastrointestinal tract and the brain, generating nutrients and modulating overall energy homeostasis. Disruption of the gut microbiota (dysbiosis) is implicated in the pathogenesis of neurodegenerative disorders of aging like AD, ALS, and HD [58,89]. Using *Caenorhabditis elegans* ALS models with impaired carnitine shuttle (transport of LCFAs across the mitochondrial membrane for energy production via βOX was defective), Labarre et al. demonstrated that *Lacticaseibacillus rhamnosus* HA-114 supplying FAs restores lipid homeostasis and energy balance. Moreover, the collected data revealed that HA-114 requires acdh-1, kat-1, and elo-6 (orthologs of human Acyl-CoA dehydrogenase short/branched chain (ACADSB), mitochondrial acetyl-CoA acetyltransferase (ACAT1) and FA elongase (ELOVL3/6)), which are associated with FA metabolism and oxidation, as the core components of this neuroprotective mechanism. However, further investigation is needed to determine whether the FAs of HA-114 exert rescuing effects directly through neuronal βOX or other pathways [75]. Triheptanoin is a triglyceride of heptanoic acid studied to treat LCFA oxidation disorders. The anaplerotic properties of triheptanoin also make it an attractive possible treatment for several neurological diseases. In animal models of ALS, epilepsy, and ischaemic stroke, it delayed motor symptoms and had a neuroprotective effect. In addition, it reduced the effort needed to undertake exercises in HD. Moreover, Adanyeguh et al. demonstrated that triheptanoin corrects the bioenergetic profile in the brain of patients with HD at an early stage of the disease [90,91].

In aging, reduced normal biological function and cognitive abilities are commonly observed in individuals without overt neurological disease. The increased FA and KBs utilization observed during aging are probably associated with a metabolic shift in astrocytes to support increased mitochondrial metabolism [92]. Using *Drosophila melanogaster* as a model system, Laranjeira et al. [80] set out to identify genes linked to the functional decline of the brain with age. The study revealed the age-dependent up-regulation of genes involved in the metabolic process of βOX in the nervous tissue of female wild-type flies. Moreover, in parallel, they demonstrated that genetic silencing of genes related to FAO improved neuronal function at a cellular and behavioral level in old flies. It was also suggested that mobilization of lipids in βOX, rather than accumulation of TAGs in the brain, is primarily responsible for the deleterious effects observed with age. However, future studies should investigate how lipid content and forms (FFAs/TAGs/LDs) change during aging and how these changes affect neuronal metabolic processes [93].

## 4. Conclusions

The brain is an organ requiring a vast quantity of lipids to maintain stable function, and FAs are the primary components in the production of almost all classes of lipids. An increasing number of studies demonstrate a significant role of βOX in maintaining global brain health, thus indicating essential metabolic connections between different brain compartments/cells.

## Figures and Tables

**Figure 1 ijms-24-13977-f001:**
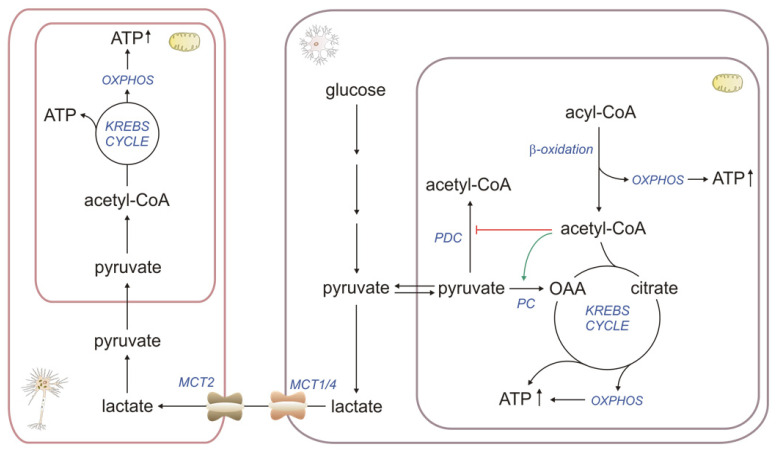
βOX as an energy source: OXPHOS—oxidative phosphorylation, PDC—pyruvate dehydrogenase complex, PC—pyruvate carboxylase, MCT—monocarboxylate transporter, ATP—adenosine triphosphate, inhibition arc is coloured in red, stimulation arc is coloured in green.

**Figure 2 ijms-24-13977-f002:**
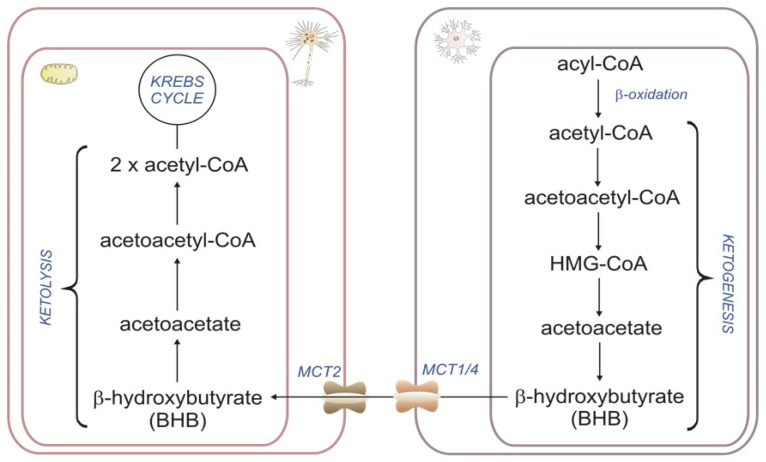
βOX as a source of ketone bodies: HMG-CoA—β-hydroxy β-methylglutaryl-CoA.

**Figure 3 ijms-24-13977-f003:**
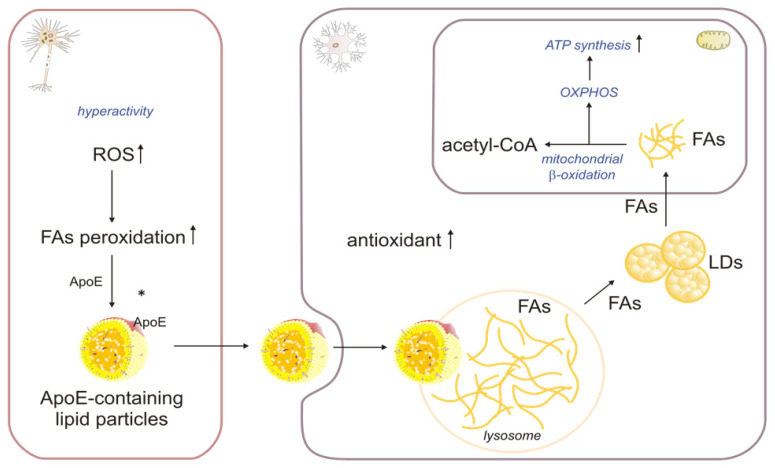
The role of βOX in the degradation of fatty acids transported from neurons to astrocytes: ROS—reactive oxygen species, LDs—lipid droplets, FAs—fatty acids. * The location of the ApoE neuronal lipidation has not been fully confirmed.

**Figure 4 ijms-24-13977-f004:**
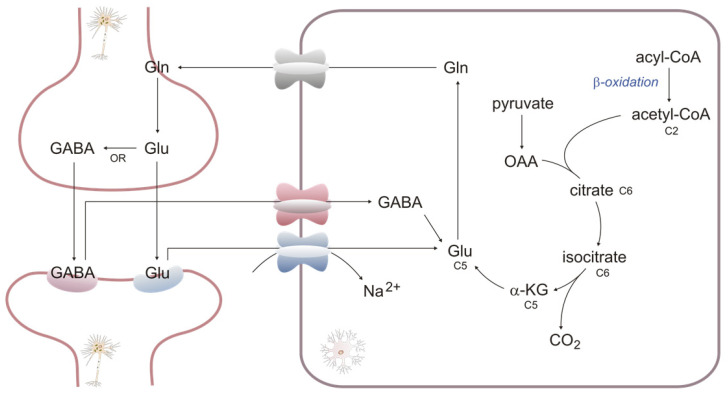
βOX as a source of carbons for neurotransmitter synthesis: Gln—glutamine, Glu—glutamic acid, GABA—gamma-aminobutyric acid, αKG—α-ketoglutarate, OAA—oxaloacetate, CO_2_—carbon dioxide.

**Figure 5 ijms-24-13977-f005:**
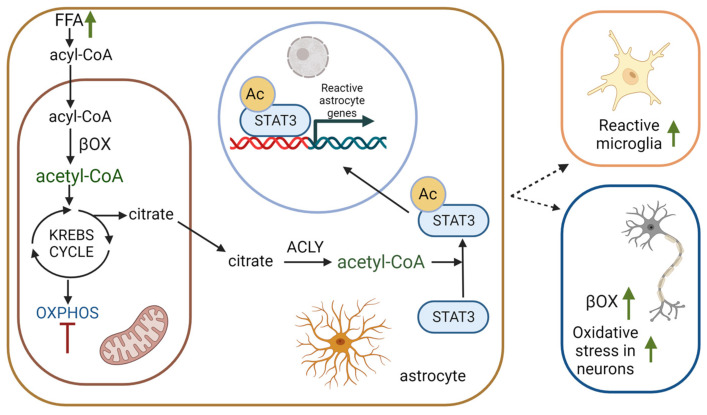
βOX as a source of acetyl-CoA in acetylation. The presence of astrocytes with an abundance of lipids induced oxidative stress, neuronal βOX, and stimulated reactive microglia formation. Ac—acetyl group, STAT3—signal transducer and activator of transcription 3, ACLY—ATP citrate lyase, inhibition arc is coloured in red, increased concentration/level is presented as green coloured arc, Created with BioRender.com.

## Data Availability

No new data were created.

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
