# Peer review of "The Role of Acyl-CoA β-Oxidation in Brain Metabolism and Neurodegenerative Diseases"

_ijms, 2023, doi:10.3390/ijms241813977_

Round 1
Reviewer 1 Report
General comments
In the present Review article, Szrok-Jurga et al summarize the roles of fatty acid beta-oxidation in normal brain metabolism and in neurodegenerative disorders. The authors have compiled an impressive amount of complex information, largely from original research findings, and written a comprehensive overview on the topic of mitochondrial b-oxidation of fatty acids in the nervous system. I have only a few suggestions for more complete coverage (otherwise specify mitochondrial Acyl-CoA b-oxidation in the title and state this omission in the Introduction:
1) From the perspective of this reviewer, the peroxisomal b-oxidation of very-long chain fatty acids (VLCFA) is completely ignored. In particular, since inherited defects in VLCFA metabolism – involving their synthesis as well as degradation – are associated with neurodegenerative disorders. Therefore, I recommend to add this important aspect to the review.
2) The focus on astrocytes is appropriate but the other neural cell types and microglia are hardly discussed, and mainly as recipients of astrocyte-generated metabolites for energy. However, at least a small section with more information on their endogenous b-oxidation potential and/or roles in providing lipids for each other would be of interest. For example, a recent paper from K.-A. Nave’s group suggest that mammalian oligodendrocytes tolerate glucose deprivation better than astrocytes and can use myelin-derived lipids as substrate for b-oxidation to provide axons with energy (Asadollahi et al 2022 bioRxiv, https://doi.org/10.1101/2022.02.24.481621).
Specific comments
1) Fig. 1: The figure is nice and helpful. However, it would be more convenient to follow, if it was split up into several figures to allow embedding of the different panels in the sections were they are actually described. This is also justified by the lack of explanatory text in the legend, which now consists of a subtitle and the abbreviations for each panel.
Fig. 1c) The figure indicates that ApoE is loaded with lipds within the Neuron. Is that intentional? According to my understanding, the lipidation of ApoE would occur extracellularly.
Fig. 1e) In panel e, the omission in the cartoon of stimulated FA b-oxidation in neurons, which is mentioned in the text, but including “decreased synaptic activity” (not discussed), seems odd.
2) Chapter 2.3, line 167-169: The statement “Unlike neurons, astrocytes make LDs, …(47)”, is not entirely correct. Also neurons can form LDs under conditions of lipid accumulation with lipotoxicity and in ageing. Suggestion: add “readily” after “astrocytes”.
3) References: At least two of the references are duplicated in the Reference list, #21 = 55 and #61 = 76. Please, fix and check the rest carefully.
Specific minor comments
Line 177: Spell out FFA (fre fatty acids).
Lines 184-185 + 189: Change “ApoE4 knocking-in” to ApoE4 knock-in.
Line 187: “the next study” is unclear here. Replace ”next” by another.
Line 237: For clarity, replace “changed” in “acyl-CoA is changed to acetyl-CoA” by “degraded” (or catabolized).
Line 298: “PAPRs” should presumably be PPARs.
Comments on Language
Largely, the article is clearly written in adequate English language. A few grammatical errors could still be improved; throughout the text singular/plural usage is often mixed up, for example in contexts like: Abstract, line 13: “role of fatty acids b-oxidation in …” where acid should be in singular (or rephrased as “b-oxidation of fatty acids”). Similarly, FA(s) oxidation, LCFA(s) transport, KB(s) levels and many more.
Specific minor language and typesetting comments
In all subheadings (2.1, 2.4, 2.5, 2. b) starting with “bOX”, the b is now part of the section numbering (2.1. b etc.).
Line 117: Delete “while”.
Line 122: Delete “respectively”.
Author Response
Dear Reviewer,
We are grateful for taking the time to review our manuscript. All valuable comments certainly helped us to improve this publication. We appreciate all suggestions which have helped us improve this publication's quality.
Yours respectfully,
Sylwia Szrok-Jurga and Ewa Stelmanska

Reviewer 2 Report
The MS discussed crucial role of astrocytic beta-oxidation in our brain. Within entire review Authors showed us very high quality article meta-analysis, so the reading of MS was very exciting. Once, what I’ve missing was the participation of peroxisomes, which recently have been showed to play a crucial role in the energy production. In the state of energy shortages, peroxisomes interacts with lipid droplets to conduct FAA shortening (by using peroxisomal FAO), which is further conduct with mitochondrial FAO. Please, include that part, so the story about astrocytic FAO will show the entire cellular network providing lipid metabolism.
Author Response
Dear Reviewer,
We are grateful for taking the time to review our manuscript. All valuable comments certainly helped us to improve this publication. We appreciate all suggestions which have helped us improve this publication's quality.
As recommended, we have added a section concerning peroxisomes' role in the oxidation of fatty acids in the brain.
Yours respectfully,
Sylwia Szrok-Jurga and Ewa Stelmańska
Reviewer 3 Report
Dear Authors,
The article is a comprehensive review of the topic covered. The authors have done a remarkable work.
Since many abbreviations are used, please make sure that the expansion of all abbreviations is given. For example: Please expand TAG in line 188.
Minor editing of the English language will be good. For example, line 64- It should be: “Reduced oxidative damage of LCFAs…..).
Minor editing of English language will be good.
Author Response
Dear Reviewer,
We are grateful for taking the time to review our manuscript. All valuable comments certainly helped us to improve this publication. We appreciate all suggestions which have helped us improve this publication's quality.
We have considered all suggestions and corrected mistakes as recommended.
Yours respectfully,
Sylwia Szrok-Jurga and Ewa Stelmanska
Round 2
Reviewer 2 Report
No further comments